# GRAPH NEURAL NETWORKS FOR REASONING 2-QUANTIFIED BOOLEAN FORMULAS

## ABSTRACT

It is valuable yet remains challenging to apply neural networks in logical reasoning tasks. Despite some successes witnessed in learning **SAT** (Boolean Satisfiability) solvers for *propositional logic* via Graph Neural Networks (GNN), there haven't been any successes in learning solvers for more complex *predicate logic*. In this paper, we target the **QBF** (Quantified Boolean Formula) satisfiability problem, the complexity of which is in-between propositional logic and predicate logic, and investigate the feasibility of learning GNN-based solvers and GNN-based heuristics for the cases with a universal-existential quantifier alternation (so-called 2QBF problems).

We conjecture, with empirical support, that GNNs have certain limitations in learning 2QBF solvers, primarily due to the inability to reason about a set of assignments. Then we show the potential of GNN-based heuristics in CEGAR-based solvers, and explore the interesting challenges to generalize them to larger problem instances. In summary, this paper provides a comprehensive surveying view of applying GNN-based embeddings to 2QBF problems, and aims to offer insights in applying machine learning tools to more complicated symbolic reasoning problems.

## 1 INTRODUCTION

As deep learning makes astonishing achievements in the domain of image (He et al., 2016) and audio (Hannun et al., 2014) processing, natural languages (Vaswani et al., 2017), and discrete heuristics decisions in games (Silver et al., 2017), there is a profound interest in applying the relevant techniques in the field of logical reasoning. Logical reasoning problems span from simple propositional logic to complex predicate logic and high-order logic, with known theoretical complexities from NP-complete (Cook, 1971) to semi-decidable and undecidable (Church, 1936). Testing the ability and limitation of machine learning tools on logical reasoning problems leads to a fundamental understanding of the boundary of learnability and robust AI, and addresses the interesting questions in decision procedures in logic, symbolic reasoning, and program analysis and verification as defined in the programming language community.

There have been some successes in learning propositional logic reasoning (Selsam et al., 2019; Amizadeh et al., 2019), which focus on **SAT** (Boolean Satisfiability) problems as defined below. A propositional logic formula is an expression composed of Boolean constants ($\top$ : true, $\bot$ : false) , Boolean variables ($x_i$), and propositional connectives such as $\wedge$, $\vee$, $\neg$ (for example $(x_1 \vee \neg x_2) \wedge (\neg x_1 \vee x_2)$). The SAT problem asks if a given formula can be satisfied (evaluated to $\top$) by assigning proper Boolean values to the variables. A crucial feature of the logical reasoning domain (as is visible in the SAT problem) is that the inputs are often structural, where logical connections between entities (variables in SAT problems) are the key information. Accordingly, previous successes have used GNN (Graph Neural Networks) and message-passing based embeddings to solve SAT problems.

However, it should be noted that logical decision procedures is more complex that just reading the formulas correctly. It is unclear if GNN embeddings (via simple message-passing) contain all the information needed to reason about complex logical questions on top of the graph structures derived from the formulas, or whether the complex embedding schemes can be learned from backpropagation. Previous successes on SAT problems argued for the power of GNN, which can handle NP-complete problems (Selsam et al., 2019; Amizadeh et al., 2019), but no successes have been reported for solving semi-decidable predicate logic problems via GNN. In order to find out where the limitation of GNN

is and why, in learning logical reasoning problems, we decide to look at problems with complexity in-between SAT and predicate logic problems, for which **QBF** (Quantified Boolean Formula) problems serve as excellent middle steps. QBF is an extension of propositional formula, which allows quantifiers ($\forall$ and $\exists$) over the Boolean variables (such as $\forall x_1 \exists x_2. (x_1 \vee \neg x_2) \wedge (\neg x_1 \vee x_2)$). In general, a quantified Boolean formula in *prenex normal form* can be expressed as such:

$$Q_i X_i Q_{i-1} X_{i-1} ... Q_0 X_0. \phi$$

where $Q_i$ are quantifiers that always differ from their neighboring quantifiers, $X_i$ are disjoint sets of Boolean variables, and $\phi$ is a propositional formula with all Boolean variables bounded in $Q_i$. Complexity-wise, QBF problems are PSPACE-complete (Kleine Büning & Bubeck, 2009), which lies in-between the NP-completeness of SAT problems and the semi-decidability of predicate logic problems. Furthermore, 2-QBF (QBF with only two alternative quantifiers) is $\Sigma_2^P$-complete (Kleine Büning & Bubeck, 2009).

Another direction of addressing logical reasoning problems via machine learning is to learn heuristic decisions within traditional decision procedures. This direction is less appealing from a theoretical perspective, but more interesting from a practical one, since it has been shown to speed up SAT solvers in practical settings (Selsam & Bjørner, 2019). In this direction, there is less concern about the embedding power of GNN, but more about the design of the training procedures (what is the data and label for training) and how to incorporate the trained models within the decision procedures. The embeddings captured via GNN is rather preferred to be lossy to prevent overfitting (Selsam & Bjørner, 2019).

In this paper we explore the potential applications of GNNs to 2QBF problems. In Section 2, we illustrate our designs of GNN architectures for embedding 2QBF formulas. In Section 3, we evaluate GNN-based 2QBF solvers, and conjecture with empirical evidences that the current GNN techniques are unable to learn *complete* SAT solvers or 2QBF solvers. In Section 4, we demonstrate the potential of our GNN-based heuristics for selecting candidates and counter-examples in the CEGAR-based solver framework. In Section 5, we discuss the related work and conclude in Section 6. Throughout the paper we redirect details to supplementary materials.

We make the following contributions:

1. Design and test possible GNN architectures for embedding 2QBF.
2. Pinpoint the limitation of GNN in learning logical decision procedures that need reasoning about a space of Boolean assignments.
3. Learn GNN-based CEGAR solver heuristics via supervised learning and uncover interesting challenges for GNN to generalize across graph structures.

## 2 GNN Embedding of Propositional Logical Formulas

**Preliminary: Graph Neural Networks.** GNNs refer to the neural architectures devised to learn the embeddings of nodes and graphs via message-passing. Resembling the generic definition in Xu et al. (2019), they consist of two successive operators to propagate the messages and evolve the embeddings over iterations:

$$m_v^{(k)} = \text{Aggregate}^{(k)}\left(\{h_u^{(k-1)} : u \in \mathcal{N}(v)\}\right), \quad h_v^{(k)} = \text{Combine}^{(k)}\left(h_v^{(k-1)}, m_v^{(k)}\right) \tag{1}$$

where $h_v^{(k)}$ denotes the hidden state (embedding) of node $v$ at the $k^{th}$ layer/iteration, and $\mathcal{N}(v)$ denotes the neighbors of node $v$. In each iteration, the $\text{Aggregate}^{(k)}(\cdot)$ aggregates hidden states from node $v$'s neighbors to produce the new message (*i.e.*, $m_v^{(k)}$) for node $v$, and $\text{Combine}^{(k)}(\cdot, \cdot)$ computes the new embedding of $v$ with its last state and its current message. After a specific number of iterations (*e.g.*, $K$), the embeddings should capture the global relational information of the nodes, which can be fed into other neural network modules for specific tasks.

**GNN Architecture for Embedding SAT formulas.** Previous success (Selsam et al., 2019) of GNN-based SAT solvers embedded SAT formulas like below. Each SAT formula is translated into a bipartite graph, where one kind of nodes represent all literals (Boolean variables and their negations, denoted as $L$), and the other kind of nodes represent all clauses (sets of literals that are connected

via $\vee$, denoted as $C$). Edges between literal and clause nodes represent the literal appearing in that clause, and all edges are represented by a sparse adjacent matrix (EdgeMatrix ($\mathbb{E}$)) of dimension $|C| \times |L|$. There is also another kind of edges connecting literals with their negations. The graph representation of $(x_1 \vee \neg x_2) \wedge (\neg x_1 \vee x_2)$ is given below as an example. Note that this architecture is specific for propositional formulas in Conjunctive Normal Form (CNF), which is composed of clauses connected via $\wedge$.

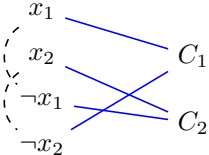

The embeddings of literals and clauses are initialized with tiled random vectors. Then the GNN uses MLPs to compute the messages of literals and clauses from the embeddings, and LSTMs to update embeddings with the aggregated messages. The mathematical process for one iteration of message-passing is given below, where $\mathrm{Emb}_L$ and $\mathrm{Emb}_C$ denotes embedding matrices of literals and clauses respectively, $\mathrm{Msg}_{X \to Y}$ denotes messages from $X$ to $Y$, $\mathrm{MLP}_X$ denotes MLPs of $X$ for generating messages from the embeddings, $\mathrm{LSTM}_X$ denotes LSTMs of $X$ for digesting incoming messages and updating the embeddings, and $\cdot$ $^T$ $[]$ represent matrix multiplication, transposition, and concatenation respectively. Furthermore, $\mathrm{Emb}_{\neg L}$ denotes a permutational view of $\mathrm{Emb}_L$ such that the same row of $\mathrm{Emb}_L$ and $\mathrm{Emb}_{\neg L}$ are embeddings of a variable and its negation respectively.

$$
\begin{aligned}
\mathrm{Msg}_{L \to C} &= \mathbb{E} \cdot \mathrm{MLP}_L(\mathrm{Emb}_L) && \text{\#aggregate clauses} \\
\mathrm{Emb}_C &= \mathrm{LSTM}_C(\mathrm{Emb}_C, \mathrm{Msg}_{L \to C}) && \text{\#combine clauses} \\
\mathrm{Msg}_{C \to L} &= \mathbb{E}^T \cdot \mathrm{MLP}_C(\mathrm{Emb}_C) && \text{\#aggregate literals} \\
\mathrm{Emb}_L &= \mathrm{LSTM}_L(\mathrm{Emb}_L, [\mathrm{Msg}_{C \to L}, \mathrm{Emb}_{\neg L}]) && \text{\#combine literals}
\end{aligned}
\tag{2}
$$

Note that different instances of MLPs and LSTMs are used for clauses and literals (they have different subscripts). What's more, $\mathrm{Emb}_{\neg L}$ is used as additional message when updating $\mathrm{Emb}_L$.

**GNN Architectures for Embedding 2QBF.** The difference between SAT formulas and 2QBF is that in 2QBF the variables are quantified by $\forall$ or $\exists$. To reflect that difference in graph representation, we separate $\forall$-literals and $\exists$-literals into different groups of nodes. For example, the graph representation of $\forall x_1 \exists x_2. (x_1 \vee \neg x_2) \wedge (\neg x_1 \vee x_2)$ is shown below:

Accordingly, in GNN architectures, the separated $\forall$-literals and $\exists$-literals are embedded via different modules. The GNN architecture design closely resembles the design philosophy of Selsam et al. (2019) in terms of permutation invariance and negation invariance, and would most likely carry over the success of GNN in solving SAT problems to 2QBF problems.

$$
\begin{aligned}
\mathrm{Msg}_{L \to C} &= [\mathbb{E}_\forall \cdot \mathrm{MLP}_\forall(\mathrm{Emb}_\forall), \mathbb{E}_\exists \cdot \mathrm{MLP}_\exists(\mathrm{Emb}_\exists)] && \text{\#aggregate clauses} \\
\mathrm{Emb}_C &= \mathrm{LSTM}_C(\mathrm{Emb}_C, \mathrm{Msg}_{L \to C}) && \text{\#combine clauses} \\
\mathrm{Msg}_{C \to \forall} &= \mathbb{E}_\forall^T \cdot \mathrm{MLP}_{C \to \forall}(\mathrm{Emb}_C) && \text{\#aggregate } \forall \\
\mathrm{Emb}_\forall &= \mathrm{LSTM}_\forall(\mathrm{Emb}_\forall, [\mathrm{Msg}_{C \to \forall}, \mathrm{Emb}_{\neg \forall}]) && \text{\#combine } \forall \\
\mathrm{Msg}_{C \to \exists} &= \mathbb{E}_\exists^T \cdot \mathrm{MLP}_{C \to \exists}(\mathrm{Emb}_C) && \text{\#aggregate } \exists \\
\mathrm{Emb}_\exists &= \mathrm{LSTM}_\exists(\mathrm{Emb}_\exists, [\mathrm{Msg}_{C \to \exists}, \mathrm{Emb}_{\neg \exists}]) && \text{\#combine } \exists
\end{aligned}
\tag{3}
$$

Note that we use $\forall$ and $\exists$ to denote all $\forall$-literals and all $\exists$-literals respectively. We use $\mathbb{E}_X$ to denote the EdgeMatrix between $X$ and $C$, and $\mathrm{MLP}_{C \to X}$ to denote MLPs that generate $\mathrm{Msg}_{C \to X}$. We de facto have tested more GNN architectures for 2QBF (see the supplementary material A.1), yet the model above performed the best in our later evaluation, so we used it in the main paper.

# 3 GNN-BASED SOLVERS FAIL IN 2QBF PROBLEMS

In the previous section, we have discussed GNN-based embeddings in propositional logical formulas. We then test whether GNN-based 2QBF solvers can be learned, following the previous successes (Selsam et al., 2019; Amizadeh et al., 2019).

## 3.1 EMPIRICAL STUDY FOR REASONING 2QBF BY GNN

**Data Preparation.** In training and testing, we follow the previous work (Chen & Interian, 2005) to generate random 2QBF formulas of *specs* (2,3) and *sizes* (8,10). That is to say, each clause has 5 literals, 2 of them are randomly chosen from a set of 8 $\forall$-quantified variables, and 3 of them are randomly chosen from a set of 10 $\exists$-quantified variables. We modify the generation procedure so that it generates clauses until the formula becomes unsatisfiable. We then randomly negate an $\exists$-quantified literal per formula to get a very similar but satisfiable formula.

**Predicting Satisfiability.** We first tested whether our graph embeddings can be used to predict satisfiability of 2QBF formulas. We extended the GNN architectures with a *voting* MLP ($\text{MLP}_{vote}$) that takes the embeddings of the $\forall$-variables after the propagation, and uses the average votes as logits for satisfiability/unsatisfiability prediction:

$$\text{logits}_{sat} = \text{mean}(\text{MLP}_{vote}(\text{Emb}_\forall))$$

We trained our GNNs with different amount of data (40 pairs, 80 pairs, and 160 pairs of satisfiable/unsatisfiable formulas) and different numbers of message-passing iterations (8 iters, 16 iters, and 32 iters), and then evaluated the converged models on 600 pairs of new instances. We report the accuracy rate of unsatisfiable and satisfiable formulas as tuples for both the training dataset and the testing dataset. By alternating the random seeds, the models with the best training data performance are selected and shown in Table 1. Since pairs of satisfiable/unsatisfiable formulas are only different by one literal, it forces the GNNs to learn subtle structural differences in the formulas. The GNNs fit well to the smaller training dataset but have trouble for 160 pairs of formulas (numbers in the green color). Performances of the models on testing dataset are close to random guesses (numbers in the blue color), and running more iterations during testing does not help with their performances.

**Predicting Unsatisfiability Witnesses.** Previous work (Selsam et al., 2019; Amizadeh et al., 2019) also showed successes in predicting *satisfiable* witnesses (variable assignments that satisfy the formulas) of SAT problems. 2QBF problems have *unsatisfiable* witnesses (assignments to $\forall$ variables that render the reduced propositional formulas unsatisfiable). Next, we test if we can train GNNs to predict unsatisfiable witnesses of 2QBF formulas. Specifically, the final embeddings of $\forall$-variables are transformed into logits via an *assignment* MLP ($\text{MLP}_{asn}$) and then used to compute the cross-entropy loss with the actual unsatisfiability witnesses of the formula:

$$\text{logits}_{witness} = \text{MLP}_{asn}(\text{Emb}_\forall)$$

Once again we tried different amount of training data (160, 320, and 640 unsatisfiable formulas) and different numbers of iterations (8 iters, 16 iters, and 32 iters), and then tested the converged models on 600 unsatisfiable new 2QBF formulas. We report the accuracy per variable and the accuracy per formula as tuples for both the training dataset and the testing dataset in Table 2, from which we can observe that the GNNs fit well to the training data (numbers in green color), especially with more message-passing iterations. However, the GNN performance on testing data is only slightly better than random guesses (numbers in blue color), and running more iterations during testing does not help with the performance either.

## 3.2 WHY GNN-BASED 2QBF SOLVER FAILED

In contrast to our initial expectation, the results above clearly show that GNNs fail to learn a 2QBF solver, unlike the previous successes in solving SAT. To investigate what limits GNNs in 2QBF solver, we first backtrack one step and examine the performance of GNNs on SAT problems.

**Difficulty in Proving Unsatisfiability for SAT problems.** Interestingly, previous works showed that GNN-based SAT solvers actually had trouble in predicting unsatisfiability with high confidence

Table 1: GNN Performance to Predict SAT/UNSAT     Table 2: GNN Performance to Predict Witnesses of UNSAT

| DATASET | 40 PAIRS | 80 PAIRS | 160 PAIRS | DATASET | 160 UNSAT | 320 UNSAT | 640 UNSAT |
|---|---|---|---|---|---|---|---|
| 8 ITERS | (0.98, 0.94) | (1.00, 0.92) | (0.84, 0.76) | 8 ITERS | (1.00, 0.99) | (0.95, 0.72) | (0.82, 0.28) |
| TESTING | (0.40, 0.64) | (0.50, 0.48) | (0.50, 0.50) | TESTING | (0.64, 0.06) | (0.67, 0.05) | (0.69, 0.05) |
| 16 ITERS | (1.00, 1.00) | (0.96, 0.96) | (0.88, 0.70) | 16 ITERS | (1.00, 1.00) | (0.98, 0.87) | (0.95, 0.69) |
| TESTING | (0.54, 0.46) | (0.52, 0.52) | (0.54, 0.48) | TESTING | (0.64, 0.05) | (0.65, 0.05) | (0.65, 0.06) |
| 32 ITERS | (1.00, 1.00) | (0.98, 0.98) | (0.84, 0.80) | 32 ITERS | (1.00, 1.00) | (0.99, 0.96) | (0.91, 0.57) |
| TESTING | (0.32, 0.68) | (0.52, 0.50) | (0.52, 0.50) | TESTING | (0.63, 0.05) | (0.64, 0.05) | (0.63, 0.05) |

(Selsam et al., 2019), if those formulas do not have a small unsatisfiable core (minimal number of clauses that is enough to cause unsatisfiability). Another work (Amizadeh et al., 2019) even completely removed unsatisfiable formulas from the training dataset (since they slowed down the training process), and only trained for predicting solutions to satisfiable formulas. However, these defects are not a problem for SAT solvers, since predicting satisfiability with high confidence is already good enough for a binary distinction.

The difficulty in proving unsatisfiability is understandable, since constructing a proof of unsatisfiability demands a complete reasoning in the search space, which is more complex than constructing a proof of satisfiability that only requires a witness. Traditionally it relies on the recursive/iterative decision procedures that either traverse all possible assignments (implicitly or explicitly) to construct the proof (DPLL (Davis et al., 1962)), or generate extra constraints from assignment trials that lead to conflicts, until some of the constraints contradict each other (CDCL (Silva et al., 2009)). In comparison, the message-passing scheme in GNN doesn't seem to resemble either of those procedures, but is rather similar to a subfamily of *incomplete* SAT solvers (WalkSAT (Selman et al., 1993)) that randomly assign variables and stochastically search for local witnesses. Similarly, those SAT solvers cannot prove unsatisfiability.

**GNN-based 2QBF Solver is Conjecturally Infeasible**   For 2QBF problems, constructions of both proofs (of satisfiability and unsatisfiability) need complete reasoning about a search space. Proof of satisfiability needs to show that for all possible assignments of $\forall$ variables, there exist satisfying assignments of $\exists$ variables, while proof of unsatisfiability needs to show that given a witness (assignment of $\forall$ variables), the reduced propositional formula is proven unsatisfiable. Given this, it's reasonable to conjecture that GNNs are probably incapable of constructing either proofs, thus being unable to learn a 2QBF solver. Traditional decision procedures (such as CEGAR-based solvers (Rabe et al., 2018)) has a way to incrementally construct such proof, but it is unlikely that the message-passing scheme in GNNs is capable of such task. Here, we conjecture that learning *complete* SAT solvers and 2QBF solvers are infeasible with the current GNN architectures and message-passing schemes.

## 4    LEARN GNN-BASED HEURISTICS FOR 2QBF

In Section 3, we conjecture (with empirical support) that GNN-based 2QBF solvers are infeasible. Thus the successes of learning GNN-based SAT solvers (Selsam et al., 2019; Amizadeh et al., 2019) cannot be simply extended to more complex logical reasoning problems. Therefore we pivot our attention to learning GNN-based heuristics that work with traditional decision procedures. Considerable QBF decision procedures have been proposed due to the importance of QBF solvers in fields such as conditional planning (Rintanen, 1999) and symbolic model checking (Plaisted et al., 2003). In this section, we will just focus on the CEGAR (Counter Example Guided Abstraction Refinement) based solving algorithm.

### 4.1    CEGAR-BASED 2QBF ALGORITHM

We first present the CEGAR-based solving procedure (Janota & Silva, 2011) in Algorithm 1. Iteratively, the CEGAR algorithm proposes an assignment of all $\forall$ variables as a *candidate*, which reduces the 2QBF formula to a SAT formula. If the SAT formula is proven unsatisfiable, the candidate becomes a *witness* and the algorithm returns (unsat, witness). Otherwise, a satisfying assignment

of $\exists$ variables can be found as a *counter-example*. Each counter-example disables a set of potential candidates, and this constraints on candidates can be expressed via *accumulated* clauses in the constraint SAT formula $\omega$ (details in supplementary material A.2). New candidates must be proposed from the satisfying solutions of $\omega$, to avoid proposing candidates that are already countered (thus *abstract refinement*). As counter-examples add clauses to $\omega$, $\omega$ may become unsatisfiable, meaning that no more candidates can be proposed. In that case, the algorithm returns (sat, -).

The algorithm is clearly exponential, since both of the search spaces (of the candidates and the counter-examples) are exponential. It is also intuitive that the *quality* of candidates and counter-examples affects the runtime of the algorithm. The traditional decision procedures have proposed a MaxSAT-based heuristics, which states that the good candidates should maximize the number of unsatisfied clauses in the formula (thus making the reduced SAT problem difficult), and the good counter-examples should maximize the number of satisfied clauses in the formula (thus providing a strong constraint on the candidates) (Janota & Silva, 2011). However, MaxSAT-based heuristics are not practical, due to the heavy overhead of MaxSAT procedures. Furthermore, the number of clauses only relates to the difficulty of the SAT problems and the strength of the constraints, but does not directly decide it. This motivates us to test whether GNN-based heuristics can be used instead.

---

**Algorithm 1** CEGAR-based 2QBF Algorithm

---

**Input:** $\forall X \exists Y \phi$
**Output:** (sat, -) or (unsat, witness)
Initialize constraints $\omega$ as an empty set of clauses.
**while** true **do**
    # proposing candidates
    (has-candidate, candidate) = SAT-solver($\omega$)
    **if** not has-candidate **then**
        **return** (sat, -)
    **end if**
    # proposing counter-examples
    (has-counter, counter) = SAT-solver($\phi_{[X \to \text{candidate}]}$)
    **if** not has-counter **then**
        **return** (unsat, candidate)
    **end if**
    # abstract refinement
    # details in supplementary material A.2
    add counter to constraints $\omega$
**end while**

---

## 4.2 BASIC SETUPS FOR GNN-BASED HEURISTICS

There are challenges in integrating neural-based heuristics into CEGAR-based solvers, since each proposed assignment (candidate or counter-example) must fit some logical constraints (i.e. they must satisfy a SAT formula). It is rather difficult to add logical constraints to neural-based proposals, but relatively easy to employ neural-based ranking on proposals that already satisfy the logical constraints. In fact, it is rather easy to ask for multiple satisfying assignments from SAT solvers, if there exist. Therefore we choose to use the GNN-based embeddings to rank multiple assignments, instead of directly predicting the best assignments. We also benefit from more training data and less risk of overfitting with the ranking methodology.

To get rankings from the GNN-based embeddings, we first transform the embeddings (of all $\forall$ variables or all $\exists$ variables) into scoring matrix (Sm) via a *scoring* MLP (MLP$_{score}$). Then a batch of assignments ($\mathbb{A}$) are ranked by passing through a two-layer perceptron (using the Sm and a learnable weighting vector Wv as weights without biases).

$$
\begin{aligned}
\text{Sm} &= \text{MLP}_{score}(\text{Emb}) \\
\text{RankingScoresLogits} &= \text{ReLU}(\mathbb{A} \cdot \text{Sm}) \cdot \text{Wv}
\end{aligned}
$$

During training, we make use of the TensorFlow ranking library (Pasumarthi et al., 2019) to compute the pairwise-logistic-loss with NDCG-lambda-weight. We then incorporate the trained models into the CEGAR cycles by replacing the SAT-solver subroutine with a procedure that returns the highest ranked solution from multiple solutions to a given SAT formula. Note that when used in CEGAR-based solvers, the GNN models only need to embed each formula once to get the scoring matrix (Sm), which is then used in all the following iterations to solve that formula. This is a significant improvement compared with previous work (Lederman et al., 2018).

The evaluations are done on 4 separate datasets:
- TrainU: 1000 unsatisfiable formulas used for training
- TrainS: 1000 satisfiable formulas used for training
- TestU: 600 unsatisfiable formulas used for testing
- TestS: 600 satisfiable formulas used for testing

| Table 3: Performance of CEGAR Candidate-Ranking | | | | | Table 4: Performance of CEGAR CounterExample-Ranking | | | | |
|---|---|---|---|---|---|---|---|---|---|

| DATASET | TRAINU | TRAINS | TESTU | TESTS | DATASET | TRAINU | TRAINS | TESTU | TESTS |
|---|---|---|---|---|---|---|---|---|---|
| - | 21.976 | 34.783 | 21.945 | 33.885 | - | 21.976 | 34.783 | 21.945 | 33.885 |
| MAXSAT | 13.144 | 30.057 | 12.453 | 28.863 | MAXSAT | 14.754 | 22.265 | 14.748 | 21.638 |
| GNN1 | 14.387 | 31.800 | 14.273 | 30.588 | GNN3 | 16.95 | 26.717 | 16.743 | 24.325 |
| GNN2 | 13.843 | 31.404 | 13.787 | 30.273 | GNN4 | 17.492 | 26.962 | 17.198 | 25.198 |

with 2 baselines:

- -: vanilla CEGAR without ranking
- MaxSAT: ranking by the number of satisfied clauses via on-the-fly formula simplification (Note that although MaxSAT performs the best in our evaluations, it is too expensive to use in practice. See asymptotic analysis in supplementary material A.2)

via measuring the average number of iterations needed to solve the 2QBF problems. Here we choose to measure the number of iterations rather than the wall clock time, because the former only measures the quality of our heuristics, while the latter is subject to various optimizations and implementation details that involve lots of engineering effort (out of the scope of this paper). From multiple random seeds, we report the results of the models that perform best on the training datasets.

## 4.3 RANKING THE CANDIDATES

Since the size of 2QBF formulas for training are quite small (the same dataset as in Section 3), we can basically enumerate all assignments in the search space to generate the training data. The interesting question left is how we assign ranking scores to all the possible candidates. One way is to follow the MaxSAT-style and rank them based on the number of clauses they satisfy (the less the better, shown as "GNN1" in Table 3). Another way is to rank them based on the number of solutions to the reduced SAT formula (the less the better, shown as "GNN2" in Table 3), since having less solutions relates to more difficult SAT problems, thus stronger candidates (see details of ranking scores in supplementary material A.2).

As shown in Table 3, all 3 ranking heuristics (including the GNN-based and the MaxSAT-baseline) improved the solving performances of all 4 datasets. The improvement on unsatisfiable problems is more significant, since we are ranking the candidates. GNN2 seems slightly better than GNN1, implying that the training data by hardness (number of solutions to the reduced SAT formula) is probably better.

## 4.4 RANKING THE COUNTEREXAMPLES

We generate the training dataset for counter-examples in a similar fashion. Once again we propose two ways to generate the ranking scores. One way is to follow the MaxSAT-style and rank them based on the number of clauses they satisfy (the more the better, shown as "GNN3" in Table 4). Another way is to adjust the ranking score of "GNN3" based on whether the counter-examples associate with the unsatisfiable cores in $\omega$, the constraint SAT formula (shown as "GNN4" in Table 4, see details of ranking scores in supplementary material A.2).

As shown in Table 4, all 3 ranking heuristics improved the solving performances of all 4 datasets. The improvement on satisfiable problems is more significant this time, since we are ranking the counter-examples. However, GNN4 performs slightly worse than GNN3. The result implies that the additional information regarding the unsatisfiable cores in 2QBF, beyond our expectation, actually damages the GNN-based heuristic. The likely explanation is that information associated with unsatisfiable cores in 2QBF may be too complicated for GNN, which goes back to the limitation of GNN in reasoning about the whole solution space and unsatisfiability.

## 4.5 COMBINATION OF THE HEURISTICS

It is reasonable to assume that ranking both the candidates and the counter-examples will further improve the solver performance. We retrained GNN models using the ranking datasets for both candidates and counter-examples, so that we still just do one GNN embedding per formula. We evaluated GNN1-3 (combining the training data of GNN1 and GNN3), GNN2-3 (combining the

training data of GNN2 and GNN3), and GNN2-4 (combining the training data of GNN2 and GNN4). As shown in Table 5, GNN2-3 is arguably our best GNN-based model via this ranking method. We further compute relative improvement of GNN2-3, which is the ratio of improvement via GNN2-3 from "-" over the improvement via MaxSAT from "-", represented by percentages. That is shown in the last row of Table 5 as GNN2-3R.

Table 5: Performance of CEGAR Both-Ranking

| DATASET | TRAINU | TRAINS | TESTU | TESTS |
|---------|--------|--------|-------|-------|
| -       | 21.976 | 34.783 | 21.945 | 33.885 |
| MAXSAT  | 9.671  | 20.777 | 9.425 | 19.883 |
| GNN1-3  | 12.505 | 25.505 | 12.22 | 24.638 |
| GNN2-3  | 11.25  | 24.76  | 12.008 | 24.295 |
| GNN2-4  | 11.686 | 25.021 | 11.605 | 24.318 |
| GNN2-3R | 87.1%  | 71.6%  | 79.4% | 68.5% |

## 4.6 EVALUATION OF LARGER 2QBF PROBLEMS

We then tested the performance of our best GNN-based heuristics (GNN2-3) on larger 2QBF problems that are extended in two different ways. On one hand, we fixed the *specs* (number of $\forall$ and $\exists$ literals per clause) but increased the *sizes* (the total number of $\forall$ variables or $\exists$ variables per formula). This essentially generated larger graphs with similar connectivity (as in the upper half of Table 6). On the other hand, we fixed the *sizes* but increased the *specs* (as in the lower half of Table 6), which essentially generated graphs with different vertex degrees. We changed the number of clauses per formula such that about half of the randomly generated 2QBF formulas are satisfiable.

We list the evaluation results in Table 6. The DataSet column shows the *specs* (the first tuple), the *sizes* (the second tuple), and the satisfiability status with the number of clauses per formula (the letter/number after the second tuple).

Table 6: Performance on Larger 2QBF

| DATASET | - | MAXSAT | GNN2-3 | GNN2-3R |
|---------|------|--------|--------|---------|
| (2,3)(16,20)U188 | 289.84 | 110.19 | 157.80 | 73.4% |
| (2,3)(16,20)S188 | 569.39 | 218.14 | 335.78 | 66.5% |
| (2,3)(8,40)U521  | 74.125 | 28.388 | 42.875 | 68.3% |
| (2,3)(8,40)S521  | 238.30 | 223.93 | 232.20 | 42.4% |
| (3,3)(8,10)U200  | 26.625 | 9.857  | 18.027 | 51.2% |
| (3,3)(8,10)S200  | 49.838 | 31.863 | 42.639 | 40.1% |
| (2,4)(8,10)U262  | 26.723 | 10.538 | 17.369 | 57.8% |
| (2,4)(8,10)S262  | 45.817 | 28.023 | 37.163 | 48.6% |
| (3,4)(8,10)U510  | 36.6   | 14.196 | 35.265 | 6.0%  |
| (3,4)(8,10)S510  | 71.846 | 48.088 | 71.992 | -0.6% |

As shown in Table 6, the GNN-based heuristics generalizes well to larger *sizes* (the upper half of Table 6). The relative improvement via GNN2-3 is about 70% compared with that of the MaxSAT baseline (modulo the (2,3)(8,40)S521 dataset which is hard to improve with either heuristics for some reasons), which is similar to its performance on smaller instances in Table 5. On the other hand, the GNN-based heuristics cannot generalize so well to instances with larger *specs*. For the dataset with either one more $\forall$-literal per clause, or one more $\exists$-literal per clause, the relative improvement via GNN2-3 is about 50%. For the dataset with both one more $\forall$-literal and one more $\exists$-literal per clause, the GNN2-3 failed completely in generalization.

This reveals an interesting challenge in GNN-based embedding or structural data embedding in general. It is natural for GNN-based embedding to generalize to larger graphs if the vertex degrees remain unchanged. In that case, it is almost like embedding a larger batch of data. However, it is not intuitive to claim that the GNN-based embedding generalizes to graphs with different vertex degrees. This caveat should promote more researches on message-passing schemes and structural data embedding in general.

# 5 Related Work and Discussion

Without using the existing decision procedures, several reasoning methods purely based on neural networks are proposed for SAT solvers. Selsam et al. (2019) presented a GNN architecture that embedded the propositional formulas. From single bit supervision (the formula is satisfiable or not), the GNN learned a procedure to find satisfying assignments before issuing predictions. Also, the GNN embeddings converge given more embedding iterations, indicating that the learned procedure is stable. Amizadeh et al. (2019) further improved this line of work by adapting a RL-style explore-exploit mechanism but considering circuit-SAT problems and DAG embeddings. They trained their DAG architectures via guided gradient descent and showed that their DAG embeddings found solutions faster than the previous GNN-embeddings, but didn't even try to tackle unsatisfiable formulas. Our paper tries to extend them to 2QBF problems, and we show that the inability to reason about unsatisfiability prevent GNNs to be a 2QBF solver. Recent work of Xu et al. (2019) discussed GNN's expressivity power, but not in the logical reasoning context.

Samulowitz & Memisevic (2007) applied classification to predict optimal choices of heuristics inside a portfolio-based and a dynamic QBF solver . Similar to our work, (Lederman et al., 2018) targeted the 2QBF problem and used GNN-based embeddings to learn branching heuristics in CADET solver in a reinforcement learning setting. However, they have to embed an updated 2QBF formula for each branching step, thus incurring high embedding overhead. To reduce the overhead, the authors used very simple GNN architectures. They also used a small number of message-passing iterations (in fact, one iteration performed best), which defeats the purpose of GNN, because 1-iteration GNN reduces to a neighbor-counting model. On the contrary, we design our solver/heuristics such that only one GNN embedding is needed per formula, which significantly reduces the GNN inference overhead. As a result, we can use more sophisticated GNN architectures with more message-passing iterations.

Belief propagation (BP) is a Bayesian message-passing method first proposed by Pearl (1982), which is a useful approximation algorithm and has been applied to the SAT problems (specifically in 3-SAT (Mézard et al., 2002)) and 2QBF problems (Zhang et al., 2012). BP can find the witnesses of unsatisfiability of 2QBF by adopting a bias estimation strategy. Each round of BP allows the user to select the most biased $\forall$-variable and assign the biased value to the variable. After all the $\forall$-variables are assigned, the formula is simplified by the assignment and sent to SAT solvers. The procedure returns the assignment as a witness of unsatisfiability if the simplified formula is unsatisfiable, or UNKNOWN otherwise. However, the fact that BP is used for each $\forall$-variable assignment leads to high overhead, similar to the RL approach given by (Lederman et al., 2018). It is interesting, however, to see that with the added overhead, BP can find witnesses of unsatisfiability, which is what one-shot GNN-based embeddings cannot achieve.

QBF problems attracted lots of research attentions due to its theoretical interests and practical applications in artificial intelligence (Rintanen, 1999), automated theorem proving (Ranjan et al., 2004), and sequential circuit verification (Sheeran et al., 2000). The subclass of 2QBF is worthy of studying in its own rights, due to applications in AI planning generalized to non-deterministic domains (Rabe et al., 2018), and planning with exponentially long plans (PSPACE-complete) (Castellini et al., 2001).

# 6 Conclusion

In this paper we investigated GNN-based 2QBF solvers and GNN-based 2QBF heuristics. We revealed the previously unrecognized limitation of GNN in reasoning about unsatisfiability of SAT problems, and conjectured that this limitation prevents GNN from learning solvers for more complex logical reasoning problems such as 2QBF satisfiability problem. This limitation is probably rooted in the simpility of message-passing scheme, which is good enough for embedding graph structures, but not for conducting complex reasoning on top of the graph structures. We then demonstrated that learning GNN-based 2QBF heuristics is potentially successful, though still faces interesting challenges in terms of generalization across graph structures. Our work extends previous progress in this field, and offers insights in applying machine learning tools to symbolic reasoning in general.

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

# A APPENDIX

## A.1 ALL GNN-EMBEDDING ARCHITECTURES

We use subscript symbols $\forall$ to denote all $\forall$-quantified literals, $\exists$ to denote all $\exists$-quantified literals, $L$ to denote all literals, and $C$ to denote all clauses. We use notations $\text{Emb}_X$ to denote embeddings of $X$, where $X$ can be subscript $\forall$, $\exists$, $L$, or $C$. We use notations $\text{Emb}_{\neg X}$ to denote embeddings of the negations of $X$ ($\forall$, $\exists$, or $L$), which is permutational views of $\text{Emb}_X$ such that the same row of $\text{Emb}_X$ and $\text{Emb}_{\neg X}$ are embeddings of a variable and its negation respectively. We use notations $\text{Msg}_{X \to Y}$ to denote messages from $X$ to $Y$. We also use notations $\text{MLP}_X$ to denote MLPs that generate messages from the embeddings of $X$, notations $\text{MLP}_{X \to Y}$ to denote MLPs that generate messages from the embeddings of $X$ for $Y$, notations $\text{LSTM}_X$ to denote LSTMs that update embeddings of $X$ given incoming messages, and notations $\text{LSTM}_{X \leftarrow Y}$ to denote LSTMs that update embeddings of $X$ given incoming messages from $Y$. We also use notations $\mathbb{E}_X$ to denote the sparse adjacency matrix of $X$ ($\forall$, $\exists$, or $L$) and clauses, notations $X \cdot Y$ to denote matrix multiplication of $X$ and $Y$, notations $[X, Y]$ to denote matrix concatenation of $X$ and $Y$, and notations $X^T$ to denote matrix transposition of $X$.

Our first form of GNN message-passing of 2QBF (Model 1) is given below.

**Model 1:**

$$\text{Msg}_C = \mathbb{E}_\forall \cdot \text{MLP}_\forall(\text{Emb}_\forall) + \mathbb{E}_\exists \cdot \text{MLP}_\exists(\text{Emb}_\exists)$$
$$\text{Emb}_C = \text{LSTM}_C(\text{Emb}_C, \text{Msg}_C)$$
$$\text{Msg}_{C \to \forall} = \mathbb{E}_\forall^T \cdot \text{MLP}_C(\text{Emb}_C)$$
$$\text{Emb}_\forall = \text{LSTM}_\forall(\text{Emb}_\forall, [\text{Msg}_{C \to \forall}, \text{Emb}_{\neg \forall}])$$
$$\text{Msg}_{C \to \exists} = \mathbb{E}_\exists^T \cdot \text{MLP}_C(\text{Emb}_C)$$
$$\text{Emb}_\exists = \text{LSTM}_\exists(\text{Emb}_\exists, [\text{Msg}_{C \to \exists}, \text{Emb}_{\neg \exists}])$$

In Model 2, we update the clause embedding with 2 LSTMs, each of them take the messages from $\forall$ and $\exists$ literals respectively.

**Model 2:**

$$\text{Msg}_{\forall \to C} = \mathbb{E}_\forall \cdot \text{MLP}_\forall(\text{Emb}_\forall)$$
$$\text{Msg}_{\exists \to C} = \mathbb{E}_\exists \cdot \text{MLP}_\exists(\text{Emb}_\exists)$$
$$\text{Emb}_C = \text{LSTM}_{C \leftarrow \forall}(\text{Emb}_C, \text{Msg}_{\forall \to C})$$
$$\text{Emb}_C = \text{LSTM}_{C \leftarrow \exists}(\text{Emb}_C, \text{Msg}_{\exists \to C})$$
$$\text{Msg}_{C \to \forall} = \mathbb{E}_\forall^T \cdot \text{MLP}_C(\text{Emb}_C)$$
$$\text{Emb}_\forall = \text{LSTM}_\forall(\text{Emb}_\forall, [\text{Msg}_{C \to \forall}, \text{Emb}_{\neg \forall}])$$
$$\text{Msg}_{C \to \exists} = \mathbb{E}_\exists^T \cdot \text{MLP}_C(\text{Emb}_C)$$
$$\text{Emb}_\exists = \text{LSTM}_\exists(\text{Emb}_\exists, [\text{Msg}_{C \to \exists}, \text{Emb}_{\neg \exists}])$$

We switch the order of these 2 LSTMs in Model 3.

**Model 3:**

$$\text{Msg}_{\exists \to C} = \mathbb{E}_\exists \cdot \text{MLP}_\exists(\text{Emb}_\exists)$$
$$\text{Msg}_{\forall \to C} = \mathbb{E}_\forall \cdot \text{MLP}_\forall(\text{Emb}_\forall)$$
$$\text{Emb}_C = \text{LSTM}_{C \leftarrow \exists}(\text{Emb}_C, \text{Msg}_{\exists \to C})$$
$$\text{Emb}_C = \text{LSTM}_{C \leftarrow \forall}(\text{Emb}_C, \text{Msg}_{\forall \to C})$$
$$\text{Msg}_{C \to \forall} = \mathbb{E}_\forall^T \cdot \text{MLP}_C(\text{Emb}_C)$$
$$\text{Emb}_\forall = \text{LSTM}_\forall(\text{Emb}_\forall, [\text{Msg}_{C \to \forall}, \text{Emb}_{\neg \forall}])$$
$$\text{Msg}_{C \to \exists} = \mathbb{E}_\exists^T \cdot \text{MLP}_C(\text{Emb}_C)$$
$$\text{Emb}_\exists = \text{LSTM}_\exists(\text{Emb}_\exists, [\text{Msg}_{C \to \exists}, \text{Emb}_{\neg \exists}])$$

In Model 4 we concatenate the messages from $\forall$ and $\exists$ literals.

**Model 4:**

$$\mathrm{Msg}_C = [\mathbb{E}_\forall \cdot \mathrm{MLP}_\forall(\mathrm{Emb}_\forall), \mathbb{E}_\exists \cdot \mathrm{MLP}_\exists(\mathrm{Emb}_\exists)]$$
$$\mathrm{Emb}_C = \mathrm{LSTM}_C(\mathrm{Emb}_C, \mathrm{Msg}_C)$$

$$\mathrm{Msg}_{C \to \forall} = \mathbb{E}_\forall^T \cdot \mathrm{MLP}_C(\mathrm{Emb}_C)$$
$$\mathrm{Emb}_\forall = \mathrm{LSTM}_\forall(\mathrm{Emb}_\forall, [\mathrm{Msg}_{C \to \forall}, \mathrm{Emb}_{\neg \forall}])$$

$$\mathrm{Msg}_{C \to \exists} = \mathbb{E}_\exists^T \cdot \mathrm{MLP}_C(\mathrm{Emb}_C)$$
$$\mathrm{Emb}_\exists = \mathrm{LSTM}_\exists(\mathrm{Emb}_\exists, [\mathrm{Msg}_{C \to \exists}, \mathrm{Emb}_{\neg \exists}])$$

The performance of our GNN architectures improve greatly after we realize that (in Model 5) we may also need to use different MLP modules to generate messages from clauses to $\forall$ and $\exists$ literals. Note that this is also the model we reported in the main paper, and the model we decided to use for all results reported in the main paper.

**Model 5:**

$$\mathrm{Msg}_C = [\mathbb{E}_\forall \cdot \mathrm{MLP}_\forall(\mathrm{Emb}_\forall), \mathbb{E}_\exists \cdot \mathrm{MLP}_\exists(\mathrm{Emb}_\exists)]$$
$$\mathrm{Emb}_C = \mathrm{LSTM}_C(\mathrm{Emb}_C, \mathrm{Msg}_C)$$

$$\mathrm{Msg}_{C \to \forall} = \mathbb{E}_\forall^T \cdot \mathrm{MLP}_{C \to \forall}(\mathrm{Emb}_C)$$
$$\mathrm{Emb}_\forall = \mathrm{LSTM}_\forall(\mathrm{Emb}_\forall, [\mathrm{Msg}_{C \to \forall}, \mathrm{Emb}_{\neg \forall}])$$

$$\mathrm{Msg}_{C \to \exists} = \mathbb{E}_\exists^T \cdot \mathrm{MLP}_{C \to \exists}(\mathrm{Emb}_C)$$
$$\mathrm{Emb}_\exists = \mathrm{LSTM}_\exists(\mathrm{Emb}_\exists, [\mathrm{Msg}_{C \to \exists}, \mathrm{Emb}_{\neg \exists}])$$

We also explore the possibility (in Model 6) of having two embeddings for each clause, one serving the $\forall$ literals and one serving the $\exists$ literals. We need extra notations: $\mathrm{Emb}_{X \to Y}$ denotes embeddings of $X$ that serves $Y$. $\mathrm{LSTM}_{X \to Y}$ denotes LSTMs that updates embedding of $X$ that serves $Y$.

**Model 6:**

$$\mathrm{Msg}_C = [\mathbb{E}_\forall \cdot \mathrm{MLP}_\forall(\mathrm{Emb}_\forall), \mathbb{E}_\exists \cdot \mathrm{MLP}_\exists(\mathrm{Emb}_\exists)]$$
$$\mathrm{Emb}_{C \to \forall} = \mathrm{LSTM}_{C \to \forall}(\mathrm{Emb}_{C \to \forall}, \mathrm{Msg}_C)$$
$$\mathrm{Emb}_{C \to \exists} = \mathrm{LSTM}_{C \to \exists}(\mathrm{Emb}_{C \to \exists}, \mathrm{Msg}_C)$$

$$\mathrm{Msg}_{C \to \forall} = \mathbb{E}_\forall^T \cdot \mathrm{MLP}_{C \to \forall}(\mathrm{Emb}_{C \to \forall})$$
$$\mathrm{Emb}_\forall = \mathrm{LSTM}_\forall(\mathrm{Emb}_\forall, [\mathrm{Msg}_{C \to \forall}, \mathrm{Emb}_{\neg \forall}])$$

$$\mathrm{Msg}_{C \to \exists} = \mathbb{E}_\exists^T \cdot \mathrm{MLP}_{C \to \exists}(\mathrm{Emb}_{C \to \exists})$$
$$\mathrm{Emb}_\exists = \mathrm{LSTM}_\exists(\mathrm{Emb}_\exists, [\mathrm{Msg}_{C \to \exists}, \mathrm{Emb}_{\neg \exists}])$$

We further explore possibility (in Model 7) that our embedding scheme should reflect a CEGAR cycle, which starts from $\forall$ variables (proposing candidates), to clauses, to $\exists$ variables (finding counterexamples), back to clauses, then back to $\forall$ variables.

**Model 7:**

$$\text{Msg}_{\forall \to C} = \mathbb{E}_\forall \cdot \text{MLP}_\forall(\text{Emb}_\forall)$$

$$\text{Emb}_{C \to \exists} = \text{LSTM}_{C \to \exists}(\text{Emb}_{C \to \exists}, \text{Msg}_{\forall \to C})$$

$$\text{Msg}_{C \to \exists} = \mathbb{E}_\exists^T \cdot \text{MLP}_{C \to \exists}(\text{Emb}_{C \to \exists})$$

$$\text{Emb}_\exists = \text{LSTM}_\exists(\text{Emb}_\exists, [\text{Msg}_{C \to \exists}, \text{Emb}_{\neg \exists}])$$

$$\text{Msg}_{\exists \to C} = \mathbb{E}_\exists \cdot \text{MLP}_\exists(\text{Emb}_\exists)$$

$$\text{Emb}_{C \to \forall} = \text{LSTM}_{C \to \forall}(\text{Emb}_{C \to \forall}, \text{Msg}_{\exists \to C})$$

$$\text{Msg}_{C \to \forall} = \mathbb{E}_\forall^T \cdot \text{MLP}_{C \to \forall}(\text{Emb}_{C \to \forall})$$

$$\text{Emb}_\forall = \text{LSTM}_\forall(\text{Emb}_\forall, [\text{Msg}_{C \to \forall}, \text{Emb}_{\neg \forall}])$$

## A.2 CEGAR Algorithm and Ranking Scores

**Steps of Abstract-Refinement in CEGAR-based 2QBF sovlers** This paragraph explains in detail about how abstract refinement in CEGAR-based 2QBF sovlers works (Janota & Silva, 2011) for our 2QBF formulas in CNF (conjunction normal form). Basically, abstract refinement is about maintaining and augmenting the constraint SAT formula $\omega$, the solutions of which are the candidates in the next round of iteration.

In the main paper, we said that we initialize $\omega$ as empty set of clauses. That was a simplification. Actually, we initialize $\omega$ with many variables and clauses. The variables include all the $\forall$ variables in the 2QBF formula, and a fresh variable $z_c$ for each clause $c$ in the 2QBF formula. Intuitively, the variable $z_c$ represents that the clause $c$ is not satisfied by the candidates. The $\omega$ is also initialized with many 2-sized clauses as below: for each clause $c$ in the 2QBF formula, we add clause $(\neg z_c \vee \neg l)$ for each $\forall$ literal $l$ in $c$. It should be clear that this initialization poses no constraints to all $\forall$ variables, since we can set all $z_c$ to false to satisfy all clauses in $\omega$ trivially.

For each counter example (assignment to all $\exists$ variables), we compute the set of 2QBF clauses that are not satisfied by the counter example (call them *residual clauses*). Intuitively, the constraint should say: at least one of the residual clauses should not be satisfied by the next proposed candidate, so that the current counter example cannot counter it. That constraint is realized by adding to $\omega$ one clause ($\vee z_c$ for all $c$ in the residual clauses). This clause guarantees that at least one of the residual clauses is not satisfied by the next candidates, and transfers the constraints to related $\forall$ variables via the corresponding $(\neg z_c \vee \neg l)$ clauses in $\omega$.

**Asymptotic analysis of GNN-based heuristics v.s. MaxSAT-baseline** To rigorously compare the overhead of GNN-based heuristics with the MaxSAT-baseline, let us assume a 2QBF instance with $N_\forall$ $\forall$-variables, $N_\exists$ $\exists$ variables, and $M$ clauses. We also assume that on average, each clause has $n_\forall$ $\forall$-literals and $n_\exists$ $\exists$-literals. Now we need to rank $K$ candidates or counter-examples.

The time complexity of MaxSAT-baseline (on-the-fly formula simplification) is $O(KMn_\forall)$, which is for each candidate and each clause, check if the candidate satisfies the clause in $n_\forall$ steps.

We also assume that the second dimension of scoring matrix (Sm) is $d$. The time complexity of GNN-based heuristics is equivalent to the 2 matrix multiplications of dimensions $(K, N_\forall) \times (N_\forall, d)$ and $(K, d) \times (d, 1)$, which is $O(KN_\forall d)$.

To compare the complexity, we can safely assume that both $d$ and $n_\forall$ are small constants. However, $N_\forall$ (number of $\forall$ variables in the formula) is often much smaller than $M$ (number of clauses in a formula). Moreover, in practice, matrix multiplications can be easily parallelized and accelerated on many kinds of hardware with BLAS libraries. Of course, GNN-based heuristics needs the GNN-embeddings via message-passing, but that is computed only once per formula, with the cost amortized.

**GNN1 Candidate Ranking Score:** i.e. based on a list of the numbers of satisfied clauses.

```
def n_clauses_list_2_ranking_scores(n_clauses_list):
    n_clauses_min = min(n_clauses_list)
    return [max(1, 10 - n_clauses + n_clauses_min)
            for n_clauses in n_clauses_list]
```

**GNN2 Candidate Ranking Score:** i.e. based on a list of the numbers of solutions to the simplified SAT formula.

```python
def n_solutions_2_ranking_score(n_solutions):
    if n_solutions <= 3: return 10.0 - n_solutions
    if n_solutions <= 5: return 6.0
    if n_solutions <= 8: return 5.0
    if n_solutions <= 12: return 4.0
    if n_solutions <= 16: return 3.0
    if n_solutions <= 21: return 2.0
    else: return 1.0

def n_solutions_list_2_ranking_scores(n_solutions_list):
    return [n_solutions_2_ranking_score(n_solutions) for n_solutions in n_solutions_list]
```

**GNN3 Counter-example Ranking Score:** i.e. based on a list of the numbers of satisfied clauses.

```python
def n_clauses_list_2_ranking_scores_counter(n_clauses_list):
    n_clauses_max = max(n_clauses_list)
    return [max(1, 10 - n_clauses_max + n_clauses)
            for n_clauses in n_clauses_list]
```

**GNN4 Counter-example ranking Score:** i.e. adjusted from GNN3 ranking scores based on unsatisfiable cores.

```python
def unsat_core_2_ranking_scores_counter(core_index, n_clauses_list):
    # core_index marks the index of scores that are in the unsatisfiable cores.
    n_clauses_max = max(n_clauses_list)
    scores = [max(1, 8 - n_clauses_max + n_clauses)
              for n_clauses in n_clauses_list]
    scores = numpy.array(scores)
    scores[core_index] = 10
    return scores.tolist
```

**Procedure to determine which counter-examples associate with the unsatisfiable cores:** Since each counter-example will add a clause to the constraint SAT formula $\omega$, determining the unsatisfiable cores (the smallest subset of clauses that constraints the formula to be unsatisfiable) of $\omega$ will give us the sets of counter-examples that directly associate with the unsatisfiable cores.

For satisfiable 2QBF formulas in the training dataset, we generate all clauses of $\omega$ from all counter-examples, and then solve $\omega$ with hmucSAT (Nadel et al., 2013) for unsatisfiable cores. For unsatisfiable 2QBF formulas, we again collect all clauses of $\omega$ from all counter-examples. In this case $\omega$ is satisfiable, and the solutions to it are actually witnesses of unsatisfiability. To obtain unsatisfiable cores, we add the solutions back to $\omega$ as additional constraints, until $\omega$ is unsatisfiable with the cores returned.