# OpenReview forum: "Graph Neural Networks for Reasoning 2-Quantified Boolean Formulas"
_ICLR.cc/2020/Conference — Reject_

### Official Review · AnonReviewer2 · 2019-10-18
**Official Blind Review #2**

**Rating:** 6

**Review:**

This paper first presents GNN architectures to solve 2-QBFs. They show that similar GNN architectures which work for propositional logic do not transfer to 2-QBFs, and provide some explanation for the result. Finally, they show how GNN modules can be used to speed up existing 2-QBF solvers instead. I mostly like the paper. The claims of the paper are well presented and empirically validated. Here are some suggestions / complaints:

1. It will be good to have more overview of 2-QBFs and existing solvers. How popular is CEGAR and why the improvements to its performance is important ?
2. Dataset sizes are tiny compared to current standards. Is there a reason to use such small datasets ? From the results, I don't think any of the conclusions will change significantly from the dataset size, but still its better to use larger datasets.
3. I think all tables should be self-explanatory. There are too many dataset splits and captions of tables do not provide any information.

**Experience Assessment:**

I do not know much about this area.

**Review Assessment: Checking Correctness Of Derivations And Theory:**

I assessed the sensibility of the derivations and theory.

**Review Assessment: Checking Correctness Of Experiments:**

I carefully checked the experiments.

**Review Assessment: Thoroughness In Paper Reading:**

I read the paper thoroughly.

---

### Official Review · AnonReviewer1 · 2019-10-23
**Official Blind Review #1**

**Rating:** 8

**Review:**

This paper explores how graph neural networks can be applied to test satisfiability of 2QBF logical formulas. They show that a straightforward extension of a GNN-based SAT solver to 2QBF fails to outperform random chance, and argue that this is because proving either satisfiability or unsatisfiability of 2QBF requires reasoning over exponential sets of assignments. Instead, they show that GNNs can be useful as a heuristic candidate- or counterexample- ranking model which improves the efficiency of the CEGAR algorithm for solving 2QBF.

This is a clear, well-written, and well-structured paper, and I support accepting it to ICLR. That being said, I am not as familiar with the literature on neural solvers for logic problems, so I base my review on the content within the paper more than its context in the field.

I can’t find much to fault with the writing and arguments. The GNN architecture for 2QBF (Section 2) is simple, elegant, and well-motivated as a minimal extension of successful SAT solvers. The arguments in Section 3 are convincing, and make a good case for why an algorithm such as CEGAR is necessary. Finally, the metrics in Section 4 are clearly interpretable and well-justified.

A couple questions and concerns:
In Section 3, The amount of training data (up to 160 pairs of formulas for predicting satisfiability) seems to be very small for a machine learning problem. By comparison, Selsam et al. 2019 says they train their GNN SAT solver on “millions of problems” (Section 5). Is there a good reason for using a much smaller dataset, given that 2QBF is a harder class of problem?
Section 4.2: how are the TraunU, TrainS, TestU, and TestS datasets generated?
In Section 4.6, are the models re-trained on these new distributions, or on the data described in Section 4.2? (If the latter, how does the GNN perform if re-trained on the larger-spec data?)

And minor points on clarity:
* “-” for the baseline seems a bit awkward; consider spelling out “vanilla”?
* Are all the numbers in the tables iteration counts, unless specified otherwise? It would help to restate this in the captions. Similarly, I wonder if there could be more informative names for GNN1, GNN2, GNN3, and GNN4?


**Experience Assessment:**

I do not know much about this area.

**Review Assessment: Checking Correctness Of Derivations And Theory:**

I assessed the sensibility of the derivations and theory.

**Review Assessment: Checking Correctness Of Experiments:**

I assessed the sensibility of the experiments.

**Review Assessment: Thoroughness In Paper Reading:**

I read the paper at least twice and used my best judgement in assessing the paper.

---

### Official Review · AnonReviewer3 · 2019-10-28
**Official Blind Review #3**

**Rating:** 3

**Review:**

This paper investigated the GNN-based solvers for the 2-Quantified Boolean Formula satisfiability problem. This paper points out that GNN has limitations in reasoning about unsatisfiability of SAT problems possibly due to the simple message-passing scheme. To extend the GNN-based SAT solvers to 2-QBF solvers, this paper then turns to learn GNN-based heuristics that work with traditional decision procedure, and proposes a CEGAR-based 2QBF algorithm.

Overall, the topic of combining logic reasoning and graph neural networks is interesting. But it is not clear how important is the targeted 2-QBF problem, except for testing and finding the limitations of GNN. In other words, this paper picks up a specific class of model for a very specific class of problem, which lacks sufficient and convincing motivations. Although GNN achieves success in solving SAT problems, it is not necessary that GNN is a must for solving the 2-QBF problems. Also, when analyzing the limitations of GNN, the paper makes conjecture only based on empirical results. It would be much more insightful to provide some theoretical analysis so that the paper can inspire other researchers working on different problems. Based on the above reasons, I would like to recommend a weak reject for this paper.

**Experience Assessment:**

I have published one or two papers in this area.

**Review Assessment: Checking Correctness Of Derivations And Theory:**

I assessed the sensibility of the derivations and theory.

**Review Assessment: Checking Correctness Of Experiments:**

I carefully checked the experiments.

**Review Assessment: Thoroughness In Paper Reading:**

I read the paper at least twice and used my best judgement in assessing the paper.

---

### Decision · Program_Chairs · 2019-12-19

**Decision:**

Reject

**Comment:**

This work investigates the use of graph NNs for solving 2QBF . The authors provide empirical evidence that for this type of satisfiability decision problem, GNNs are not able to provide solutions and claim this is due to the message passing mechanism that cannot afford for complex reasoning. Finally, the authors propose a number of heuristics that extend GNNs and show that these improve their performance.

2-QBF problem is used as a playground since, as the authors also point, their complexity is in between  that of predicate and propositional logic. This on its own is not bad,  as it can be used as a minimal environment for the type of investigation the authors are interested. That being said, I find a number a number of flaws in the current form of the paper (some of them pointed by R3 as well), with the main issue being that of lack experimental rigor. Given the restricted set of problems the authors consider, I think the experiments on identifying pathologies of GNNs on this setup could have gone more in depth. Let me be specific.

1) The bad performance is attributed to message-passing. However, this feels anecdotal at the moment and authors do not provide firm conclusions about that. The only evidence they provide is that performance becomes better with more message-passing iterations they allow. This is a hint though to dive deeper rather than a firm conclusion. For example do we know if the finding about sensitivity to  message-passing  is due to the small size of the network or the training procedure?
2) To add on that, there is virtually no information on the paper about the specifics of the experimental setup, so the reader cannot be convinced that the negative results do not arise from a bad experimental configuration (e.g., small size of network).
3) Moreover, the negative results here, as the authors point, seem to contradict previous work, providing negative results against GNNs.  Again, this is a valuable contribution if that is indeed the case, but again the paper does not provide enough evidence. In lieu of a convincing set of experiments, the paper could provide a proof (as also asked by R3). However with no proof and not strong empirical evidence that this result does not feel ready to get published at ICLR.

Overall, I think this paper with a bit more rigor could be a very good submission for a later conference. However, as it stands I cannot recommend acceptance.